# Environmental Factors That Affect the Sanitary and Nutritional Variability of Raw Milk in Dual Purpose Livestock Systems of Colombian Orinoquia

**DOI:** 10.3390/ani13081385

**Published:** 2023-04-18

**Authors:** Mauricio Vélez-Terranova, Rómulo Campos Gaona, Arcesio Salamanca-Carreño, Ricardo Andrés Velasco Daza, Brandon Alexis Arenas Rodríguez, José Sebastián Chaparro Ortegón

**Affiliations:** 1Facultad de Ciencias Agropecuarias, Universidad Nacional de Colombia, Palmira 763531, Colombia; 2Facultad de Medicina Veterinaria y Zootecnia, Universidad Cooperativa de Colombia, Villavicencio 500001, Colombia; 3Unidad de Docencia y Formación, Universidad Nacional de Colombia Sede Orinoquia, Arauca 810001, Colombia

**Keywords:** chemical composition, dual purpose systems, health status, low tropics

## Abstract

**Simple Summary:**

Milk quality can be affected by several factors and management conditions. The aim was to identify factors influencing the raw milk composition and sanitary quality in dual-purpose systems of the Colombian Orinoquia. Milk chemical composition was analyzed from samples obtained in 30 farms with manual milking. To study the udder health status, the California Mastitis Test (CMT) was applied to 300 cows at the time of milking. The study was carried out in the dry and rainy season. It was observed that the total daily milk production of the farm and the season influenced the compositional quality of the milk. The highest levels of protein, lactose, solid non-fat (SNF), and density were observed in the farms with a daily milk production lower than 100 kg/day. In the rainy season, milk quality was higher. The CMT test indicated that only 7.6% of mammary quarters had two or more degrees of positivity. This low value indicates that the presence of subclinical mastitis is not common in the dual-purpose systems of the evaluated region.

**Abstract:**

Milk is the natural food with the highest biological quality for the human population and its production can be affected by several sanitary factors and management conditions. With the objective of identifying influence factors on milk compositional and sanitary quality in a region with wide productive potential in the Colombian Orinoquia, an experiment was carried out in two contrasting climatic seasons. For the milk compositional analysis, samples of daily production from 30 dual-purpose systems were analyzed. Similarly, the udder sanitary status of 300 cows was studied using the California Mastitis Test (CMT). Data analysis included mixed models, Pearson correlations, frequency tables, and the Kruskal–Wallis test. The results showed that the total daily milk production of the farm and the season influenced the milk compositional quality. The farms with milk productions lower than 100 kg/day presented the highest levels of protein, lactose, solid non-fat (SNF), and density, while in the rainy season, the milk quality was higher compared to the dry season. The CMT test indicated that only 7.6% of the evaluated mammary quarters presented two or more degrees of positivity. There is an opportunity to improve the milk compositional quality by improving the nutritional offer for animals during the year. The low CMT positivity indicates that, in the calf-at-foot milking system, the presence of subclinical mastitis is not a determining variable in milk production.

## 1. Introduction

The Orinoquia region is considered the last great agricultural frontier and food pantry in Colombia. The region’s economy has revolved around extensive livestock [1]. In this region, several municipalities are part of development programs implemented by the Colombian government with the aim to generate sustainable productive projects that allow the involvement of human resources with great needs originating in the social conflict of the country [2].

Milk production in Colombia is generated from specialized dairy and dual-purpose systems (milk and meat). The latter is of great importance for the country since it is estimated that they contribute to about 60% of the national milk production [3]. Arauca department is a tropical region located in the Colombian east, where livestock production is one of the main economic activities [4]. In the zone, the dual-purpose systems have shown considerable growth because of the introduction of specialized milk production breeds, which are used in crosses with zebu animals to generate multiracial genotypes (*Bos indicus* × *Bos taurus*) that are managed extensively, with diets of low to medium quality [3,4]. Previous characterization studies of livestock production systems in the region showed that of 775 evaluated farms, 449 (58%) were dedicated to dual purposes [5]. The Arauca department reported a livestock inventory of 1,302,248 animals, of which 563,184 are adult females, with a potential milk production that can reach 500,000 L/day; this production is mainly concentrated in the Arauquita, Saravena, Fortul, and Tame municipalities under piedmont conditions [6]. The product is obtained mainly by small and medium producers and is intended for family consumption or for commercialization in collection centers or local cheesemakers [7]. Milk is considered one of the most important foods in the human diet, due to its contribution of macro and micronutrients (protein, fat, carbohydrates, minerals, vitamins, etc.), essential for adequate mental and physical development [8]. Its hygienic production and volumes that allow competitiveness require technical support and specific studies that allow the development of the dairy production activity in the territory. Compositional and microbiological milk characteristics vary depending on several factors, such as animal breed, diet, and hygienic and processing conditions during its production. It is known that the type of milking, the absence of refrigeration, and the limited sanitary control, reduce the nutritional quality and increase microorganism proliferation that can affect consumer health [9,10].

In the Colombian tropics, the chemical composition and microbiological quality reported for dual-purpose systems range between 3–3.4, 4.1–5.4, 8.1–8.7, 12.2–13.6% for protein, fat, solid non-fat, and total solids, respectively, while the somatic cells ranged among 50,000–885,000/mL [11,12]. Similarly, it has been reported that milk quality in these systems can be affected by several factors such as the animal’s breed, type of milking, and the implementation of nutritional supplementation practices [13]. In Colombia dual-purpose systems are heterogeneous, and the milk quality could be affected in different ways. Given the absence of studies on milk quality in the Arauca region, it is necessary to develop research related to the quality of raw milk and the factors that affect it, in such a way that they allow determining productive and sanitary limitations, to establish management strategies consistent with the productive environment, similar to what was conducted in other regions of the country [14]. The aim of the present study was to identify factors influencing the chemical and sanitary quality of raw milk from dual-purpose systems of the Arauquita municipality (Colombia) during different seasons of the year.

## 2. Materials and Methods

### 2.1. Study Site

From the registry of the Arauca Livestock Association (ASOGANADEROS), 30 representative dual-purpose farms from the Panama region of Arauquita (Colombia) were selected. The reference geographic coordinates are 6.823 N, 71.261 W, and the general conditions include an average altitude of 153 m above sea level, a rain period: April–November, and, drought: December–March, with average annual rainfall of 1216 mm [15]. The experiment was carried out during the year 2021.

Lactating and healthy dairy cows were chosen for the evaluation. The cattle breed component corresponded to crossbreed animals including mainly the Gyr, Holstein, Brahman, Girolando, Brown Swiss, Normande, Guzera, and Simmental breeds in different proportions. Dairy cows’ nutrition was based on rotational semi-extensive grazing mainly with *Brachiaria humidicola* and *Brachiaria mutica* pastures, mineral salt, and ad libitum water supplied. In the herds, an average of 30 milking cows (range 11–50) were found, and the milking system was manual, once a day and with the presence of the calf. Cows’ reproduction was based on natural mating or artificial insemination.

### 2.2. Compositional and Sanitary Evaluation of Raw Milk

To evaluate the compositional and microbiological quality of raw milk, 250 mL of previously homogenized milk was taken from the containers used to store the daily production on each farm. Samples were taken weekly (for four weeks) in the dry and rainy seasons. The samples were transported under refrigeration to the Bovine laboratory of the National University of Colombia—Orinoquia for the physicochemical composition analysis (fat, protein, lactose, non-fatty solids, minerals, total solids, and density), using an ultrasonic milk analyzer—Boeco^®^ LAC-SA-90 model (Hamburg, Germany). For the microbiological evaluation (colony-forming units—CFU), the SimPlate (TPC) technique was used [16]. Likewise, information related to the milking characteristics of each farm was collected, with the objective of evaluating its influence on milk composition. The evaluated variables included: farm predominant racial group, season, milking facilities conditions (paddock or corral), good milking practices implementation (handling and hygiene practices for obtaining and storing milk, udder cleaning and disinfection, milk refrigeration, udder sealing after milking, among others) and daily total milk production of the farm.

To evaluate the udder clinical status, related to the possible presence of subclinical mastitis, the California Mastitis Test (CMT) was applied. During the milking of each farm, five lactating cows were randomly selected and the CMT test was performed on each mammary quarter. The test was applied to 150 cows during the rainy and dry seasons of the area, constituting a total of 300 evaluated animals. For results interpretation, the generalized qualification of four degrees was used. Negative to frankly positive results were identified with mathematical signs based on negativity or degree of presence [17]. Additionally, information related to the season, racial group of the evaluated animal, third of lactation, calving number, body score, and individual daily milk production was collected to evaluate its influence on the total number of CMT-positive quarters. In this case, results equal to or greater than two crosses (++ or +++) were counted as a positive CMT result, which suggests a degree of inflammation and the presence of defense white cells, a situation compatible with subclinical mastitis [17].

### 2.3. Statistical Analysis

The study was carried out under a randomized complete block design, being the farm the blocking factor. The milk composition data were checked to identify and eliminate outliers, followed by descriptive statistical analysis.

The influence of variables related to farm characteristics on milk production and composition variables (total daily milk production per farm, fat, protein, fat–protein ratio, lactose, solid non-fat, minerals, total solids, density, colony forming units) was evaluated using the following mixed model:Y_ijklmn_ = μ + TM_i_ + S_j_ + RG_k_ + GMP_l_ + MF_m_ + F_n_ + ε_ijklmn_
where Y_ijk_ represents the milk production and composition variable; μ represents the general average of the observed variable; TM_i_ represents the fixed effect of “it” total farm daily milk production (≤60, 61–100, >100); S_j_ represents the fixed effect of “jt” season (rainy or dry); RG_k_ represents the fixed effect of “kt” predominant racial group of the farm (F1, cross-breed), GMP_l_ represents the fixed effect of “lt” good milking practices (none, one or two or more practices), MF_m_ represents the fixed effect of “mt” milking facilities (paddock or corral), F_n_ represents the farm random effect and ε_ijklmn_ represents the random error term. It was assumed that F_n_ and ε_ijklmn_ were independent and distributed ~N (0, s^2^). The colony forming unit’s variable was transformed to Log10 for analysis. The least significant difference (LSD) was used for mean differentiation (*p* < 0.05). Likewise, Spearman correlations were estimated between chemical and microbiological composition variables (*p* < 0.05).

Finally, the data obtained from the CMT test were analyzed using frequency tables. Subsequently, the effect of season (rainy or dry), animal racial group (F1 or cross-breed), third of lactation, calving number (1, 2 or >3), body score (≤3, 3.1 to 3.5, >3.5), individual daily milk production (1.2 to 3.5, 3.6 to 5, >5 kg/day), and the combination between third of lactation and individual milk production on the number of CMT-positive quarters were evaluated, using the Kruskal–Wallis nonparametric test. When the test was significant, pairwise comparisons were performed for the differentiation of modalities or levels of the studied effects (*p* < 0.05). All analyzes were performed using the Infostat software 2020 (Universidad Nacional de Córdoba, Córdoba, Argentina) [18].

## 3. Results

### 3.1. Chemical Composition of the Raw Milk

Among the farms, the total daily milk production varied between 50 to 143 kg/day. The mean values of the chemical and microbiological composition variables of the analyzed milk samples during the evaluated seasons are shown in Table 1.

Representative variables such as fat, protein, lactose, and TS presented mean values of 3.61, 3.36, 4.80, and 12.59%, respectively, while the log CFU attained a mean value of 4.72 (52,840.7 CFU/mL). The coefficient of variation ranged between 2.38 to 12.55% suggesting that variability was effectively controlled.

### 3.2. Influence Factors on Raw Milk Quality

The effect of factors influencing the milk quality is shown in Table 2. Total daily milk production per farm influenced the protein, lactose, SNF, and density variables (*p* < 0.05). Farms with total daily milk volumes lower than 100 kg presented higher protein, lactose, and SNF levels than farms whose daily milk production was higher than 100 kg/day (3.37 vs. 3.33, 4.85 vs. 4.79 and 9.05 vs. 8.92% for protein, lactose, and SNF, respectively). In statistical terms, the density was higher in the farms with milk volumes between 61–100 kg/day than those with volumes >100 kg/day (1033.2 vs. 1032.8 g/cm^3^). The season factor influenced most of the milk chemical composition variables, including fat, protein, F–P ratio, lactose, SNF, minerals, and TS (*p* < 0.05). Except for lactose, all the other variables presented mean values higher in the rainy season than in the dry period. The fat, protein, F–P ratio, SNF, minerals, and TS were 3.67 vs. 3.47%, 3.37 vs. 3.34%, 1.07 vs. 1.03, 9.04 vs. 8.97%, 0.75 vs. 0.73%, 12.66 vs. 12.36% for the rainy and dry season, respectively. The lactose content was 5.81 vs. 4.84% in the dry and rainy seasons, respectively.

In the study, the effects of a racial group, implementation of good milking practices, and milking facilities on the milk chemical and microbiological composition variables were non-significant (*p* > 0.05).

### 3.3. Association between Milk Quality Variables

The associations between milk quality variables are shown in Table 3. In general, several significant correlations were found (*p* < 0.05). The TS variable was positively related to the fat, protein, lactose, SNF, F–P ratio, and minerals percentages, with values between 0.50–0.94. Other significant correlations were found between protein and lactose, minerals, and SNF content (r = 0.73–0.99). Similarly, the minerals levels related to lactose, fat, F–P ratio, and SNF (r = 0.54–0.79), lactose was positively associated with SNF (r = 0.94), as well as fat and F–P ratio (r = 0.96). Finally, density was directly associated with protein (0.85), lactose (0.94), SNF (0.85), and inversely with the F–P ratio (−0.49).

### 3.4. Incidence of Subclinical Mastitis

The evaluation of the presence of subclinical inflammation in the mammary gland in response to CMT is shown in Table 4. In total, 300 cows and 1200 mammary quarters were evaluated. It was found that 71.1% of the mammary quarters were negative for CMT, 21.3% presented trace levels and the remaining 7.6% had positive results (++ or +++). Average CMT positivity per mammary quarter was similar with values ranging between 7.6–8%. Of the studied cows, 229 (76%) presented negative or trace CMT results, while the remaining animals (24%) showed one or more quarters with some degree of positivity. These results indicate that the degree of mammary gland affectation in the analyzed dual-purpose systems was low, with only 7.6% of the quarters with two or more degrees of positivity.

The factors influencing the number of CMT-positive quarters are shown in Table 5.

Only the season, third of lactation, and the variable resulting from the combination between a third of lactation and individual milk production significantly affect the CMT-positive quarters (*p <* 0.05). In general, a greater positivity was observed during the rainy season and in cows finishing lactation (last third) and with low production levels (1.2 to 3.5 kg/day).

## 4. Discussion

### 4.1. Raw Milk Composition

The observed milk quality values are like other reports obtained in tropical conditions [11,19]. Fat and protein values of 3.7 and 3.3%, respectively, have been found under low tropical conditions, with CFU below 500.000/mL [3]. During the descriptive analysis, it was observed that in the rainy and dry seasons of the evaluated systems, 50 and 26.7%, respectively, of the farms presented fat and protein percentages equal to or higher than the values reported for the low tropics. This evidence a high variability in milk quality due to the season factor, causing 50 to 73.3% of the farms to present low fat and protein values during the year. This behavior can be attributed to the variability associated with the availability and quality of the forages used for grazing, especially during the dry season (December–March) when the water deficit severely limits forage biomass production and quality [20,21] prevented from meeting the animal’s nutritional requirements, affecting the milk volume and quality in terms of fat and protein levels [22]. In this sense, the implementation of nutritional alternatives or supplementation practices during the critical seasons has shown to be useful to increase milk production and maintain its solids composition during the year [23].

Regarding the CFU, it was observed that, in the rainy and dry seasons, 33.3 and 23.3%, respectively, of the farms presented values higher than 300.000 CFU/mL, which is the limit allowed for the quality bonuses payment for the production systems located in the studied region [24]. Other studies have reported higher total bacteria counts in tank milk samples during wet and rainy seasons, mainly attributed to the greater difficulty in handling animals with a high presence of sludge in the mammary gland and extremities during milking, and to the increases in the probability of making milk handling errors under these conditions [25,26]. Despite the observed variation in the CFU during the evaluated seasons, between 66.7 to 76.7% of the farms showed CFU values lower than 300.000 CFU/mL, which suggests that within the evaluated systems, the post-milking management does not imply a significant risk on the product sanitary quality, however, there are opportunities for improvement, so it is necessary to re-evaluate the methods of obtaining and storing the milk [19].

### 4.2. Correlations between Milk Quality Variables

The results of the association milk quality variables were similar to those found in crossbred dairy cattle [27,28]. The direct relationship between TS and fat, protein, lactose, SNF, F–P ratio, and minerals percentages are expected since fat, protein, lactose, and minerals make up the total milk solids. On the other hand, SNF is made up of the same components except for fat [29]. In this way, the greater the TS content, the greater proportion of its constituents is expected, and a better opportunity for transformation in dairy by-products.

The positive association of density with protein (0.85), lactose (0.94), SNF (0.85), and negatively with the F–P ratio (−0.49) agree with what was reported in another study [28]. The same authors found a negative correlation between density and fat of −0.16, like the non-significant value obtained in the present study of −0.24. These results allow explaining the negative correlation found with the F–P index, in which fat is a component of its construction.

### 4.3. Influence Factors on Milk Quality

The results showed that in farms with total daily milk production volumes <100 kg, higher levels of protein, lactose, and SNF were found compared to farms with higher daily productivity. This is an expected effect given the negative correlation between milk production and total solids component [10,29]. Even though statistical differences were found in milk density levels between farms with daily production volumes of 61–100 and >100 kg/day (1033.2 vs. 1032.8 kg/m^3^), these values are similar to the density reported in milk samples by four processing plants located in Northwestern Colombia [30] and are within the range considered normal.

The season factor influenced most of the milk chemical composition variables (fat, protein, F–P ratio, SNF, minerals, and TS), presenting higher values during the rainy season compared to the dry period. Under subtropical conditions, it was found that milk fat and protein levels were higher during the rainy season with mean values of 4.02 and 3.36%, respectively [25]. Similarly, it has been observed that in Holstein cows exposed to stressful temperature and humidity indices > 78 (equivalent to an average of 33.8 °C and 36% of maximum ambient temperature and minimum relative humidity, respectively), milk components such as fat, protein, lactose, TS, and SNF were significantly reduced [31]. In lowland tropical conditions, it has been shown that climatic variables such as mean temperature and solar radiation increased the tympanic temperature of tropical dairy cows, reducing their performance with values above 38.9 °C [32].

Warm conditions such as that of the study area can generate heat stress cases in grazing animals that affect forage intake and the pastures’ nutritional quality that serve as the main food source in the evaluated livestock systems, resulting in a general reduction in the milk production volume and the total solids content [22,26]. Decreased feed intake due to heat stress reduces the energy available for milk production, similarly, it reduces milk protein levels by decreasing the caseins number and concentration [33], while lactose can be reduced because the supply of glucogenic agents to the mammary gland is affected. Depression of milk fat, protein, and lactose considerably reduces TS under heat stress conditions [31]. Other factors influencing could be related to inappropriate handling and storage methods, which under season with high temperatures accelerate product degradation [19]. This scenario stimulates a review of the methods of obtaining, handling, and storing the product in the evaluated farms, which together with supplementation practices with nutritional alternatives (especially in the dry season) lead to guarantee the maintenance of the milk compositional quality throughout the year.

Considering that in the present study, no differences were identified between racial groups, implementation of good milking practices, and milking facilities on the milk quality characteristics, it can be affirmed that the genetic component and milking management were similar within the dual-purpose farms in the evaluated region. To corroborate these results, it is recommended to carry out similar studies with a larger sample size.

### 4.4. Subclinical Mastitis Positivity

Individual milk production in the sampled animals during the rainy season varied on average between 2.84–6.80 kg/day, like that obtained in the dry season, ranging between 2.92–7.56 kg/day. These values are higher than the 2 to 2.6 kg/day found in different locations of the studied region [4].

The average 7.6 to 8% of CMT positivity per mammary quarter, agrees with the 9.54% positivity reported in the Colombian low tropic conditions [30]. The general 7.6% of CMT positivity indicates that the affectation degree of the mammary gland in the analyzed dual-purpose systems was low. The values found in this study are like the 11.6% of CMT positivity reported in 15 dual-purpose farms in Colombia [34], however, they are lower than the 45.9% and 54.6% prevalence found in dual-purpose farms located in the same evaluated region [35] or in other Latin American countries under tropical conditions [36]. The low CMT positivity would indicate that the milking type, which is carried out in the presence of the calf, seems to positively affect mammary gland health. The calves removed a greater quantity of milk from the mammary gland given its suction capacity and contribute to preventing the proliferation of microorganisms due to the bactericidal effect of its saliva [37]. Moreover, the neurophysiological mechanisms involved in the cow–calf bond eliciting for a more rapid and full udder emptying, making the substrate for bacterial proliferation unavailable [38].

Despite the low CMT positivity, some effects such as the season, third of lactation, and the variable resulting from the combination between a third of lactation and individual milk production, influenced the results (*p* < 0.05).

A greater positivity was observed during the rainy season, which agrees with another study reporting that mammary gland infections are accentuated under high rainfall conditions, especially due to infection with coliforms [39]. On the other hand, the cows finishing lactation (last third) presented higher cases of positivity than animals starting lactation (first third). The same trend was observed when relating the level of individual productivity with a third of lactation. In general, cows from the last third of lactation and with low production levels (1.2 to 3.5 kg/day) presented a higher frequency of CMT-positive cases compared to cows from the first third of lactation and production levels >5 kg/day. These results are consistent with what was found in crossbred Holstein cows where the microbiological quality of the milk was worse in finishing lactation animals, possibly attributed to a greater pathogens susceptibility during this phase [40]. Another possible explanation for the higher CMT positivity observed in the final lactation phases is related to the mammary gland cell proliferation and replacement dynamics during this period [41]. After the peak of milk production, the mammary gland undergoes a gradual regression through the apoptotic cell death process, indicating that considerable mammary cell renewal occurs during the final phases of lactation [42]. The foregoing suggests that apoptotic somatic cells are partly eliminated through the milk, generating a high frequency of positivity results in tests such as CMT, however, these results would be false positives since the cell turnover process during the final stages of lactation is a normal physiological response in various animal species [42]. The relationship between CMT positivity and mammary gland cell turnover requires further investigation.

Although the season, third of lactation, and third of lactation—individual milk production effects—were shown to be influential factors on CMT positivity, their effects are not representative given that in general terms, the positivity was low in the evaluated systems (7.6%).

## 5. Conclusions

The milk compositional quality in the dual-purpose systems of the Arauquita municipality presented average values of 3.61, 3.36, 4.80, 9.02, 0.75, and 12.59% for fat, protein, lactose, SNF, minerals, and TS, respectively, while the CFU average was 52,480.7/mL. The total daily milk production of the farm and the season influenced the milk compositional quality. The farms with milk productions <100 kg/day presented the highest protein, lactose, SNF, and density levels, while in the rainy season, the milk quality was higher compared to the dry period, in terms of fat, protein, F–P ratio, SNF, density, minerals, and TS. There is an opportunity to improve the milk compositional quality and its persistence over time through the implementation of practices that guarantee a greater availability and nutritional quality of the pastures destined for animal feeding and through proper milk management during the collection, handling, and storage process. The CMT results indicated that the affectation degree of the mammary gland in terms of subclinical mastitis in the evaluated dual-purpose systems was low, with only 7.6% of the quarters with two or more positivity degrees. It is recommended to continue with this type of evaluation including a larger animal number.

## Figures and Tables

**Table 1 animals-13-01385-t001:** Descriptive statistics of the raw milk composition variables obtained in the evaluated dual-purpose systems.

Variables	Mean	SD	CV (%)	Min	Max
Fat (%)	3.61	0.40	11.67	3.09	4.02
Protein (%)	3.36	0.08	2.48	3.28	3.49
F–P	1.08	0.12	11.27	0.94	1.23
Lactose (%)	4.80	0.12	2.38	4.62	4.96
SNF (%)	9.02	0.25	2.75	8.79	9.37
Minerals (%)	0.75	0.03	3.39	0.71	0.78
TS (%)	12.59	0.46	3.59	11.84	13.14
Density (kg/m^3^)	1032.95	0.88	2.63	1031.80	1034.15
Log CFU (mL)	4.72	0.59	12.55	4.28	5.07

SD: standard deviation; CV: coefficient of variation; Min: minimum; Max: maximum; F–P: fat–protein ratio; SNF: solid non-fat; TS: total solids; CFU: colony forming units.

**Table 2 animals-13-01385-t002:** Mean values of the studied effects on the milk chemical and microbiological quality in dual-purpose systems of the Arauquita municipality.

Variable	Category	Fat (%)	Protein (%)	F–P	Lactose (%)	SNF (%)	Minerals (%)	TS (%)	Density (g/cm^3^)	CFU/mL	
										Log_10_	Antilog
TDMPF (kg/day)	≤60	3.54 ± 0.10	3.37 ± 0.03 a	1.05 ± 0.03	4.85 ± 0.04 a	9.05 ± 0.08 a	0.75 ± 0.01	12.54 ± 0.14	1033.1± 0.29 ab	5.23 ± 0.17	169,824
	61–100	3.49 ± 0.10	3.37 ± 0.03 a	1.04 ± 0.03	4.85 ± 0.04 a	9.04 ± 0.08 a	0.74 ±0.01	12.50 ± 0.014	1033.2 ± 0.28 a	5.14 ± 0.16	138,038
	>100	3.58 ± 0.12	3.33 ± 0.03 b	1.07 ± 0.03	4.79 ± 0.04 b	8.92 ± 0.08 b	0.74 ± 0.01	12.49 ± 0.16	1032.8 ± 0.32 b	5.40 ± 0.19	251,189
	*p*-value	NS	0.0392	NS	0.0255	0.0206	NS	NS	0.0423	NS	
Season	Rainy	3.67 ± 0.09 a	3.37 ± 0.03 a	1.07 ± 0.03 a	4.84 ± 0.04 a	9.04 ± 0.07 a	0.75 ± 0.05 a	12.66 ± 0.14 a	1033.03 ± 0.28	5.28 ± 0.16	190,546
	Dry	3.47 ± 0.10 b	3.34 ± 0.03 b	1.03 ± 0.03 b	5.81 ± 0.04 b	8.97 ± 0.08 b	0.73 ± 0.04 b	12.36 ± 0.14 b	1033.02 ± 0.28	5.23 ± 0.16	169,824
	*p*-value	0.0106	0.0110	0.0207	0.0300	0.0068	0.0017	<0.0001	NS	NS	
Racial group	F1	3.57 ± 0.14	3.36 ± 0.04	1.06 ± 0.04	4.83 ± 0.06	9.02 ± 0.11	0.75 ± 0.01	12.55 ± 0.20	1033.1 ± 0.41	5.27 ± 0.23	186,209
	Tri-cross	3.51 ± 0.12	3.34 ± 0.04	1.05 ± 0.03	4.81 ± 0.05	8.97 ± 0.10	0.74 ± 0.01	12.46 ± 0.18	1032.9 ± 0.36	5.09 ± 0.21	123,027
	Cross-bread	3.53 ± 0.10	3.36 ± 0.03	1.05 ± 0.03	4.84 ± 0.04	9.04 ± 0.08	0.75 ± 0.01	12.53 ± 0.14	1033.1 ± 0.29	5.41 ± 0.17	257,040
	*p*-value	NS	NS	NS	NS	NS	NS	NS	NS	NS	
Good milking practices	None	3.64 ± 0.12	3.36 ± 0.03	1.08 ± 0.03	4.82 ± 0.05	9.03 ± 0.09	0.75 ± 0.01	12.63 ± 0.17	1032.0 ± 0.35	5.32 ± 0.20	208,930
	1	3.39 ± 0.09	3.36 ± 0.03	1.01 ± 0.03	4.85 ± 0.04	9.03 ± 0.07	0.74 ± 0.01	12.39 ± 0.13	1033.4 ± 0.27	5.47 ± 0.15	295,121
	≥2	3.58 ± 0.16	3.34 ± 0.05	1.07 ± 0.04	4.80 ± 0.07	8.96 ± 0.13	0.75 ± 0.01	12.52 ± 0.23	1032.8 ± 0.48	4.99 ± 0.27	97,724
	*p*-value	NS	NS	NS	NS	NS	NS	NS	NS	NS	
Milking facilities	Paddock	3.51 ± 0.16	3.36 ± 0.05	1.04 ± 0.04	4.84 ± 0.07	9.02 ± 0.13	0.74 ± 0.01	12.51 ± 0.23	1033.1 ± 0.47	5.51 ± 0.27	323,594
	Corral	3.56 ± 0.06	3.35 ± 0.02	1.02 ± 0.02	4.81 ± 0.03	8.99 ± 0.05	0.75 ± 0.003	12.51 ± 0.10	1032.9 ± 0.20	5.00 ± 0.11	100,000
	*p*-value	NS	NS	NS	NS	NS	NS	NS	NS	NS	

TDMPF: total daily milk production per farm; F–P: fat to protein ratio; SNF: solid non-fat; TS: total solids; CFU: colony forming units; NS: no significant effect. Means with different letters are different at *p* < 0.05.

**Table 3 animals-13-01385-t003:** Estimated Spearman correlations between milk quality variables in the evaluated dual-purpose systems (coefficients and their significance are below and above the diagonal, respectively).

Variable	Fat	Density	Lactose	Protein	F–P	Log CFU	Minerals	SNF	TS
Fat	1	0.2000	0.7600	0.1300	0.0000	0.6400	0.0000	0.1400	0.0000
Density	−0.24	1	0.0000	0.0000	0.0100	0.9300	0.1200	0.0000	0.2500
Lactose	0.06	0.94	1	0.0000	0.2900	0.9200	0.0021	0.0000	0.0046
Protein	0.28	0.85	0.96	1	0.9300	0.8600	0.0000	0.0000	0.0000
F–P	0.96	−0.49	−0.2	0.02	1	0.6900	0.0002	0.9300	0.0000
Log CFU	−0.09	0.02	0.02	−0.03	−0.08	1	0.7000	0.8500	0.6300
Minerals	0.79	0.29	0.54	0.73	0.62	−0.07	1	0.0000	0.0000
SNF	0.28	0.85	0.96	0.99	0.02	−0.04	0.72	1	0.0000
TS	0.89	0.22	0.5	0.68	0.74	−0.09	0.94	0.68	1

F–P: fat–protein ratio; SNF: solid non-fat; TS: total solids; CFU: colony forming units. Correlation was significant *p* < 0.05.

**Table 4 animals-13-01385-t004:** California mastitis test positivity discriminated by individual mammary quarters, within the evaluated population of dual-purpose cows.

Sanitary State	RA		LA		RP		LP		Total Quarters	
	n	%	n	%	n	%	n	%	n	%
Negative	222	74.0	216	72.0	199	66.3	216	72.0	853	71.1
+	55	18.3	63	21.0	77	25.7	61	20.3	256	21.3
++	18	6.0	16	5.3	22	7.3	19	6.3	75	6.3
+++	5	1.7	5	1.7	2	0.7	4	1.3	16	1.3
Total	300		300		300		300		1200	

RA: right anterior; LA: left anterior; RP: right posterior; LP: left posterior. +: positivity degree 1; ++: positivity degree 2; +++: positivity degree 3.

**Table 5 animals-13-01385-t005:** Factors associated with CMT positivity (++ and +++) in the analyzed sample of dual-purpose cows.

Effect	Mean	Ranks	*p*-Value
Season			
Dry	0.20	137.4 b	0.0004
Rainy	0.41	163.7 a	
Third of lactation			
1	0.23	142.0 b	0.0186
2	0.30	153.0 ab	
3	0.59	173.5 a	
Third-Individual milk			
T1-G1	0.31	154.3 ab	0.0342
T1-G2	0.27	144.9 ab	
T1-G3	0.16	134.1 b	
T2-G1	0.39	161.0 ab	
T2-G2	0.16	135.9 ab	
T2-G3	0.40	171.4 ab	
T3-G1	0.71	179.0 a	
T3-G2	0.43	167.8 ab	
T3-G3	0.50	167.7 ab	

T1, T2 y T3: third of lactation 1, 2, and 3, respectively. Individual milk: individual milk production: G1 = 1.2 a 3.5, G2 = 3.6 a 5, G3 ≥ 5 kg/día. Ranks with different letters differ statistically with *p* < 0.05.

## Data Availability

Data are available upon reasonable request to the first author.

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
