# Peer review of "Environmental Factors That Affect the Sanitary and Nutritional Variability of Raw Milk in Dual Purpose Livestock Systems of Colombian Orinoquia"

_animals, 2023, doi:10.3390/ani13081385_

Round 1

Reviewer 1 Report

Article ID animals-2304277

Dear authors,

I find your manuscript well-constructed. I've highlighted a few minor details, which I've listed below.

Good work

Specific comments

L 13: please, recast the sentence as follow “…that milk was manually milked with the presence of calves in the Colombian Orinoquia”. Thanks.

L 44: the statement “This region has been characterized” seems truncated. The region was characterized by what? Please, check the sentence arrangement, thanks.

L 59-60: Please check the correctness of the punctuation between the numerical digits. In my opinion, only the decimals should be separated. Thanks.

L 84: Please enter the name of the association in parentheses, thanks.

L 90: in my view, the term "racial" can be misunderstood. Use breed, please.

L 98-99: probably this sentence could be improved in clarity. I ask you to specify if the milk samples were collected daily (or in any case at which weekly interval) during the sampling weeks or if a single sample was collected weekly. Thanks.

L 106-106: I invite the authors to specify what they mean by "milk facilities conditions and "good milking practices". Thanks.

L 121: please, replace “Information Analysis” with “Statistical Analysis”, thanks.

L 122: to avoid unnecessary redundancy, I suggest deleting “from dual-purpose systems”, thanks.

L 125 (and along with the text): according to the journal template, “p-value” should be reported in lowercase and italics. Thanks.

L 129-134: if possible, it would be better to use cross-bread instead of "mestizo". Thanks.

L 310: In addition, the authors could underline the importance of the presence of the calf at milking on udder health by adding the following statement "Moreover, the neurophysiological mechanisms involved in the cow-calf bond eliciting for a more rapid and fully udder emptying, making the substrate for bacterial proliferation unavailable [https://doi.org/10.3390/ani11071968]". Thanks.

Author Response

First reviewer’s responses

Dear reviewer

The authors appreciate the insightful comments.

We attach all corrections and answers.

Comment

Dear authors,

I find your manuscript well-constructed. I've highlighted a few minor details, which I've listed below.

Good work

Specific comments

L 13: please, recast the sentence as follow “…that milk was manually milked with the presence of calves in the Colombian Orinoquia”. Thanks.

Response

Corrected in the text.

Comment

L 44: the statement “This region has been characterized” seems truncated. The region was characterized by what? Please, check the sentence arrangement, thanks.

Response

Corrected in the text.

Comment

L 59-60: Please check the correctness of the punctuation between the numerical digits. In my opinion, only the decimals should be separated. Thanks.

Response

Corrected in the text.

Comment

L 84: Please enter the name of the association in parentheses, thanks.

Response

Corrected in the text.

Comment

L 90: in my view, the term "racial" can be misunderstood. Use breed, please.

Response

Corrected in the text.

Comment

L 98-99: probably this sentence could be improved in clarity. I ask you to specify if the milk samples were collected daily (or in any case at which weekly interval) during the sampling weeks or if a single sample was collected weekly. Thanks.

Response

Corrected in the text.

Comment

L 106-106: I invite the authors to specify what they mean by "milk facilities conditions and "good milking practices". Thanks.

Response

The “milking facilities conditions” refer to whether they have electricity, water, floors in good condition, corrals in good condition, among others.

“Good milking practices” refer to quarter disinfection before milking, quarter sealed after milking, udder flushing, among others.

Corrected in the text.

Comment

L 121: please, replace “Information Analysis” with “Statistical Analysis”, thanks.

Response

Corrected in the text.

Comment

L 122: to avoid unnecessary redundancy, I suggest deleting “from dual-purpose systems”, thanks.

Response

Corrected in the text.

Comment

L 125 (and along with the text): according to the journal template, “p-value” should be reported in lowercase and italics. Thanks.

Response

Corrected in the text.

Comment

L 129-134: if possible, it would be better to use cross-bread instead of "mestizo". Thanks.

Response: Corrected in the text.

Comment

L 310: In addition, the authors could underline the importance of the presence of the calf at milking on udder health by adding the following statement "Moreover, the neurophysiological mechanisms involved in the cow-calf bond eliciting for a more rapid and fully udder emptying, making the substrate for bacterial proliferation unavailable [https://doi.org/10.3390/ani11071968]". Thanks.

Response

Added the statement in the text.

Reviewer 2 Report

The following comments were made to improve the manuscript.

a)     About introduction section:

·       Milk quality is influenced by genetic and environmental factors, which have not been described in the introduction.

·       It is necessary to describe previous studies about quality of milk yield under dual-purpose cattle systems.

·       It is necessary to describe previous studies about genetic and environmental factors and their effects on the milk yield and milk quality of dual-purpose cows.

·       Updated references are required.

·       It is required to appropriately write the general objective of the study.

b) About the materials and methods section:

·       It is required to describe the materials and methods used in a clear, sequential way.

·       It is necessary to describe sample size estimate.

·       It is necessary to describe the genetic, health, nutritional, and reproductive management of cows.

·       It is necessary to describe the main breeds and crossbred of cows existing in the evaluated dual-purpose production systems.

The statistical analysis section.

·       The statistical analysis carried out must be organized and described.

·       Initially describe the experimental design used, the dependent variables, as well as the independent ones in terms of their inclusion in the statistical model as fixed or random effects (their classes and number of observations).

·       Also, the linear expression of models used must be included.

·       In case of mixed models, the structure of covariances of the random effects must be described.

c) About Results and discussion section:

·       It is necessary to include Kruskal-wallis nonparametric test

·       replace +/- with ±

d) Conclusions

·       the conclusion is not a synthesis of the results, these must be raised based on the objectives.

·       references need to be checked.

Author Response

Second reviewer’s responses

Dear reviewer

The authors appreciate the insightful comments.

We attach all corrections and answers.

Comment

  About introduction section:

  • Milk quality is influenced by genetic and environmental factors, which have not been described in the introduction.

Response: In the introduction, the third paragraph specifies the factors that can affect the compositional quality of milk

“. Compositional and microbiological milk characteristics varies depending on several factors, such as animals breed, diet, and the hygienic and processing conditions during its production. It is known that the type of milking, the absence of refrigeration and the limited sanitary control, reduce the nutritional quality and increase microorganism proliferation that can affect the consumer health [9].”

Comment

  • It is necessary to describe previous studies about quality of milk yield under dual-purpose cattle systems.

Response: Corrected in the text.

Comment

  • It is necessary to describe previous studies about genetic and environmental factors and their effects on the milk yield and milk quality of dual-purpose cows.

Response: Corrected in the text.

Comment

  • Updated references are required.

Response: new and more up-to-date references were included: (10), (11), (12), (13), (21), (22), (33), (39)

Comment

  • It is required to appropriately write the general objective of the study.

Response: Another version of the objective is included in the text

Comment

  1. b) About the materials and methods section:
  • It is required to describe the materials and methods used in a clear, sequential way.

Response: the section is described sequentially according to what was done:

  1. place of study, type of animals, management and nutrition
  2. Evaluation of compositional and sanitary quality of milk
  3. Analysis of Subclinical Mastitis (CMT)

However, the section was revised to try to provide more clarity.

Comment

  • It is necessary to describe sample size estimate.

Response: The following is mentioned in the text:” From the registry of the Arauca Livestock Association ASOGANADEROS, 30 representative dual-purpose farms from the Panama region of Arauquita (Colombia) were selected.”

In this way, the sample corresponds to 30 representative production systems of dual-purpose livestock activity in the area, which included those producers willing to collaborate with the study.

Comment

  • It is necessary to describe the genetic, health, nutritional, and reproductive management of cows.

Response: the requested information is included in the text

Comment

  • It is necessary to describe the main breeds and crossbred of cows existing in the evaluated dual-purpose production systems.

Response: the requested information is included in the text

Comment

The statistical analysis section.

  • The statistical analysis carried out must be organized and described.

Response: Corrected in the text.

Comment

  • Initially describe the experimental design used, the dependent variables, as well as the independent ones in terms of their inclusion in the statistical model as fixed or random effects (their classes and number of observations).

Response: Corrected in the text.

Comment

  • Also, the linear expression of models used must be included.

Response: Corrected in the text.

Comment

  • In case of mixed models, the structure of covariances of the random effects must be described.

Response: Corrected in the text.

Comment

  1. c) About Results and discussion section:
  • It is necessary to include Kruskal-wallis nonparametric test

Response: Corrected in the text

Comment

  • replace +/- with ±

Response: Corrected in the text

Comment

  1. d) Conclusions
  • the conclusion is not a synthesis of the results, these must be raised based on the objectives.

Response: We believe that the way in which the conclusion is written responds to the objective of the study.

Comment

  • references need to be checked.

Response: the references were adjusted according to what was established by the journal

Reviewer 3 Report

This study has its merits. However, there are major concerns that must be addressed.

 General comments:

1. There are some grammatic mistakes in the manuscript and the authors should check and revise them. For example, tenses in articles should be consistent. Please revised the manuscript carefully.

2. Many statistical results were not presented in the MS.

3. Result description is not consistent with data in Table.

4. The Discussion section need to revised carefully. The Discussion section lacks logical coherence, and the subheadings do not summarize the discussion well. The distinction between the data in this article and the data reported in references is not clear.

 Specific comments:

1. line 19: (<100kg/day) is not the lowest daily milk production, revise it thorough out the MS

2. line 22-23, 228-229: suggest to add the result of nutritional supply to support this sentence. Why the author did not analyze the nutritional supply?

3. line 82: change to “2. Materials and methods”

4. 87-89: Since the author discussed the effects of temperature and humidity on milk production. They should add the climatic characteristics of rain and drought period, especially the difference in the temperature and humidity.

5. 113-114: During the milking of each farm (30 farms), five lactating cows were randomly selected. There would be 150 cows were tested with CMT. But there were 300 cows were tested. Explain it.

6. 114-117, 133-140: The result was not presented in this article.

7. line 142: The results did not show the microbiological composition, revise it.

8. 155-167: There were several mistakes, revised it carefully. For example, 157: should be F-P ratio; lin159, (r = 0.73 - 0.99) is wrong; 0,0000 is wrong. Lack the result description between chemical composition and microbiological result, since the subtitle is “Association between chemical and microbiological variables”

9. 164-167: Author should add the results of P value in Table 2.

10. 169: Should be “were”

11. 170: Keep “SNF” consistent in the MS and tale.

12. 176: “Kg/m3” is wrong

13: 179-180: Some data are wrong, please be careful. For example, “3.61 vs 3.47%” is not consistent with table 3.

14. 186-202: There were several mistakes, revised it carefully. For example, the letter lost in TDMPF factors on NFS.

15. 208: (7.6 - 8%) is not consistent with table 4.

16. 220-224: No data to support it.

17. 231-232: No data to support it.

18. 217. The subtitle cannot conclude the text.

19.270-272: So the temperature and humidity indices of rain period is lower than that of drought period? In my opinion, rain period is associated with high temperatures and humidity.

20.296-297: No data to support it.

Author Response

Third reviewer’s responses

Dear reviewer

The authors appreciate the insightful comments.

We attach all corrections and answers.

 Comment

This study has its merits. However, there are major concerns that must be addressed.

 General comments:

  1. There are some grammatic mistakes in the manuscript and the authors should check and revise them. For example, tenses in articles should be consistent. Please revised the manuscript carefully.

Response: The article was carefully checked, and errors were corrected. The text has been corrected.

English was revised according to your suggestion.

Comment

  1. Many statistical results were not presented in the MS.

Response: It was checked that the different statistical procedures carried out were described in the results section. In the same way, Table 5 was added where the results of the Kruskal Wallis test are described.

Comment

  1. Result description is not consistent with data in Table.

Response: all the results were reviewed, ensuring that they agreed with what was reported in the tables

Comment

  1. The Discussion section need to revised carefully. The Discussion section lacks logical coherence, and the subheadings do not summarize the discussion well. The distinction between the data in this article and the data reported in references is not clear.

Response: the discussion was revised for consistency. The subtitles were modified to fit the content of the text. Values are established to compare between the observed results and those reported in the literature for greater clarity.

Comment

 Specific comments:

  1. line 19: (<100kg/day) is not the lowest daily milk production, revise it thorough out the MS

Response: Corrected in the text

Comment

  1. line 22-23, 228-229: suggest to add the result of nutritional supply to support this sentence. Why the author did not analyze the nutritional supply?

Response: No nutritional quality analyzes were carried out because there were many farms and we prefer to focus on only milk analysis. Likewise, all the farms were based on grazing in a region with a monomodal rainfall regime where forage availability is known to be affected in the dry period. In this way, the statements made were supported with bibliography.

Comment

3, line 82: change to “2. Materials and methods”

Response: Corrected in the text

Comment

87-89: Since the author discussed the effects of temperature and humidity on milk production. They should add the climatic characteristics of rain and drought period, especially the difference in the temperature and humidity.

Response: Corrected in the text

Comment

  1. 113-114: During the milking of each farm (30 farms), five lactating cows were randomly selected. There would be 150 cows were tested with CMT. But there were 300 cows were tested. Explain it.

Response: CMT test was applied to 150 cows during the dry period and 150 cows during the rainy season for a total of 300 animals. The observation was included in the text.

Comment

  1. 114-117, 133-140: The result was not presented in this article.

Response: Table 5 was included, showing only the significant effects that showed an influence on the CMT positivity cases.

Comment

  1. line 142: The results did not show the microbiological composition, revise it.

Response: The CFU was the only microbiological variable that was evaluated, and this variable was not affected by any of the studied effects. In this way, the CFU is only mentioned at the beginning of the results section, indicating a description of the observed average value.

Comment

  1. 155-167: There were several mistakes, revised it carefully. For example, 157: should be F-P ratio; lin159, (r = 0.73 - 0.99) is wrong; 0,0000 is wrong. Lack the result description between chemical composition and microbiological result, since the subtitle is “Association between chemical and microbiological variables”

Response: we apologized, we had forgotten to mention in table 3, that the correlation coefficients and their significance are below and above the diagonal, respectively.

Since no correlation with the CFU variable was found, no results associated with microbiological variables are shown. In this way, and following your suggestion, the subtitle of this section was modified.

Comment

  1. 164-167: Author should add the results of P value in Table 2.

Response: Table 2, which now became Table 3, specifies that correlation coefficients and their significance are below and above the diagonal, respectively.

Comment

  1. 169: Should be “were”

Response: Corrected in the text

Comment

  1. 170: Keep “SNF” consistent in the MS and tale.

Response: Corrected in the text

Comment

  1. 176: “Kg/m3” is wrong

Response: According to the equipment instructions, the units of the density variable correspond to kg/m3. The equipment user manual can be found at the following link:

Comment

13: 179-180: Some data are wrong, please be careful. For example, “3.61 vs 3.47%” is not consistent with table 3.

Response: Corrected in the text

Comment

  1. 186-202: There were several mistakes, revised it carefully. For example, the letter lost in TDMPF factors on NFS.

Response: Corrected in the text

Comment

  1. 208: (7.6 - 8%) is not consistent with table 4.

Response: Corrected in the text

Comment

  1. 220-224: No data to support it.

Response: this information was observed during the descriptive analysis. The results are not presented because it would be necessary to build a large table with the descriptive statistics of the 30 farms, which we considered was not necessary.

Comment

  1. 231-232: No data to support it.

Response: this information was observed during the descriptive analysis. The results are not presented because it would be necessary to build a large table with the descriptive statistics of the 30 farms, which we considered was not necessary.

Comment

  1. 217. The subtitle cannot conclude the text.

Response: Corrected in the text

Comment

19.270-272: So the temperature and humidity indices of rain period is lower than that of drought period? In my opinion, rain period is associated with high temperatures and humidity.

Response: in the referenced study, an ITH >78 is equivalent to an average of 33.8 °C and 36 % of maximum ambient temperature and minimum relative humidity respectively. These conditions and other more adverse environments are easily achievable in tropical lowland conditions.

Comment

20.296-297: No data to support it.

Response: this information was observed during the descriptive analysis. We consider that just mentioning the information on individual milk production is sufficient, however, Table 5 describes the groups that were formed according to individual milk production (G1=1.2 a 3.5, G2= 3.6 a 5, G3=   >5 kg/día)

Round 2

Reviewer 2 Report

The authors made the required modifications and enhanced the manuscript.

Author Response

Dear reviewer
The authors appreciate the enlightening comments for the improvement of the manuscript.
English was corrected

Reviewer 3 Report

This study has its merits and I commend the authors for that. However, there are major concerns that must be addressed.

General comments:

1. There are some grammatic mistakes in the manuscript and the authors should check and revise them. For example, tenses in articles should be consistent. Please revised the manuscript carefully.

2. Many statistical results were not presented in the MS.

3. Result description is not consistent with data in Table.

4. The Discussion section need to revised carefully. The Discussion section lacks logical coherence, and the subheadings do not summarize the discussion well. The distinction between the data in this article and the data reported in references is not clear.

Specific comments:

1. line 19: (<100kg/day) is not the lowest daily milk production, revise it thorough out the MS

2. line 22-23, 228-229: suggest to add the result of nutritional supply to support this sentence. Why the author did not analyze the nutritional supply?

3. line 82: change to “2. Materials and methods”

4. 87-89: Since the author discussed the effects of temperature and humidity on milk production. They should add the climatic characteristics of rain and drought period, especially the difference in the temperature and humidity.

5. 113-114: During the milking of each farm (30 farms), five lactating cows were randomly selected. There would be 150 cows were tested with CMT. But there were 300 cows were tested. Explain it.

6. 114-117, 133-140: The result was not presented in this article.

7. line 142: The results did not show the microbiological composition, revise it.

8. 155-167: There were several mistakes, revised it carefully. For example, 157: should be F-P ratio; lin159, (r = 0.73 - 0.99) is wrong; 0,0000 is wrong. Lack the result description between chemical composition and microbiological result, since the subtitle is “Association between chemical and microbiological variables”

9. 164-167: Author should add the results of P value in Table 2.

10. 169: Should be “were”

11. 170: Keep “SNF” consistent in the MS and tale.

12. 176: “Kg/m3” is wrong

13: 179-180: Some data are wrong, please be careful. For example, “3.61 vs 3.47%” is not consistent with table 3.

14. 186-202: There were several mistakes, revised it carefully. For example, the letter lost in TDMPF factors on NFS.

15. 208: (7.6 - 8%) is not consistent with table 4.

16. 220-224: No data to support it.

17. 231-232: No data to support it.

18. 217. The subtitle cannot conclude the text.

19.270-272: So the temperature and humidity indices of rain period is lower than that of drought period? In my opinion, rain period is associated with high temperatures and humidity.

20.296-297: No data to support it.

Author Response

Third reviewer’s responses Round2

Dear reviewer

The authors appreciate the insightful comments.

We attach all corrections and answers.

 Comment

This study has its merits. However, there are major concerns that must be addressed.

 General comments:

  1. There are some grammatic mistakes in the manuscript and the authors should check and revise them. For example, tenses in articles should be consistent. Please revised the manuscript carefully.

Response: The article was carefully checked, and errors were corrected. The text has been corrected.

English was revised according to your suggestion.

Comment

  1. Many statistical results were not presented in the MS.

Response: It was checked that the different statistical procedures carried out were described in the results section. In the same way, Table 5 was added where the results of the Kruskal Wallis test are described.

Comment

  1. Result description is not consistent with data in Table.

Response: all the results were reviewed, ensuring that they agreed with what was reported in the tables

Comment

  1. The Discussion section need to revised carefully. The Discussion section lacks logical coherence, and the subheadings do not summarize the discussion well. The distinction between the data in this article and the data reported in references is not clear.

Response: the discussion was revised for consistency. The subtitles were modified to fit the content of the text. Values are established to compare between the observed results and those reported in the literature for greater clarity.

Comment

 Specific comments:

  1. line 19: (<100kg/day) is not the lowest daily milk production, revise it thorough out the MS

Response: Corrected in the text

Comment

  1. line 22-23, 228-229: suggest to add the result of nutritional supply to support this sentence. Why the author did not analyze the nutritional supply?

Response: No nutritional quality analyzes were carried out because there were many farms and we prefer to focus on only milk analysis. Likewise, all the farms were based on grazing in a region with a monomodal rainfall regime where forage availability is known to be affected in the dry period. In this way, the statements made were supported with bibliography.

Comment

3, line 82: change to “2. Materials and methods”

Response: Corrected in the text

Comment

87-89: Since the author discussed the effects of temperature and humidity on milk production. They should add the climatic characteristics of rain and drought period, especially the difference in the temperature and humidity.

Response: Corrected in the text

Comment

  1. 113-114: During the milking of each farm (30 farms), five lactating cows were randomly selected. There would be 150 cows were tested with CMT. But there were 300 cows were tested. Explain it.

Response: CMT test was applied to 150 cows during the dry period and 150 cows during the rainy season for a total of 300 animals. The observation was included in the text.

Comment

  1. 114-117, 133-140: The result was not presented in this article.

Response: Table 5 was included, showing only the significant effects that showed an influence on the CMT positivity cases.

Comment

  1. line 142: The results did not show the microbiological composition, revise it.

Response: The CFU was the only microbiological variable that was evaluated, and this variable was not affected by any of the studied effects. In this way, the CFU is only mentioned at the beginning of the results section, indicating a description of the observed average value.

Comment

  1. 155-167: There were several mistakes, revised it carefully. For example, 157: should be F-P ratio; lin159, (r = 0.73 - 0.99) is wrong; 0,0000 is wrong. Lack the result description between chemical composition and microbiological result, since the subtitle is “Association between chemical and microbiological variables”

Response: we apologized, we had forgotten to mention in table 3, that the correlation coefficients and their significance are below and above the diagonal, respectively.

Since no correlation with the CFU variable was found, no results associated with microbiological variables are shown. In this way, and following your suggestion, the subtitle of this section was modified.

Comment

  1. 164-167: Author should add the results of P value in Table 2.

Response: Table 2, which now became Table 3, specifies that correlation coefficients and their significance are below and above the diagonal, respectively.

Comment

  1. 169: Should be “were”

Response: Corrected in the text

Comment

  1. 170: Keep “SNF” consistent in the MS and tale.

Response: Corrected in the text

Comment

  1. 176: “Kg/m3” is wrong

Response: According to the equipment instructions, the units of the density variable correspond to kg/m3. The equipment user manual can be found at the following link:

Comment

13: 179-180: Some data are wrong, please be careful. For example, “3.61 vs 3.47%” is not consistent with table 3.

Response: Corrected in the text

Comment

  1. 186-202: There were several mistakes, revised it carefully. For example, the letter lost in TDMPF factors on NFS.

Response: Corrected in the text

Comment

  1. 208: (7.6 - 8%) is not consistent with table 4.

Response: Corrected in the text

Comment

  1. 220-224: No data to support it.

Response: this information was observed during the descriptive analysis. The results are not presented because it would be necessary to build a large table with the descriptive statistics of the 30 farms, which we considered was not necessary.

Comment

  1. 231-232: No data to support it.

Response: this information was observed during the descriptive analysis. The results are not presented because it would be necessary to build a large table with the descriptive statistics of the 30 farms, which we considered was not necessary.

Comment

  1. 217. The subtitle cannot conclude the text.

Response: Corrected in the text

Comment

19.270-272: So the temperature and humidity indices of rain period is lower than that of drought period? In my opinion, rain period is associated with high temperatures and humidity.

Response: in the referenced study, an ITH >78 is equivalent to an average of 33.8 °C and 36 % of maximum ambient temperature and minimum relative humidity respectively. These conditions and other more adverse environments are easily achievable in tropical lowland conditions.

Comment

20.296-297: No data to support it.

Response: this information was observed during the descriptive analysis. We consider that just mentioning the information on individual milk production is sufficient, however, Table 5 describes the groups that were formed according to individual milk production (G1=1.2 a 3.5, G2= 3.6 a 5, G3=   >5 kg/día)

English corrected
